# NTopo: Mesh-free Topology Optimization using Implicit Neural Representations

**Jonas Zehnder**
Department of Computer Science
and Operations Research
Université de Montréal
jonas.zehnder@umontreal.ca

**Yue Li**
Department of Computer Science
ETH Zurich
yue.li@inf.ethz.ch

**Stelian Coros**
Department of Computer Science
ETH Zurich
scoros@inf.ethz.ch

**Bernhard Thomaszewski**
Department of Computer Science
ETH Zurich
bthomasz@ethz.ch

## Abstract

Recent advances in implicit neural representations show great promise when it comes to generating numerical solutions to partial differential equations. Compared to conventional alternatives, such representations employ parameterized neural networks to define, in a mesh-free manner, signals that are highly-detailed, continuous, and fully differentiable. In this work, we present a novel machine learning approach for topology optimization—an important class of inverse problems with high-dimensional parameter spaces and highly nonlinear objective landscapes. To effectively leverage neural representations in the context of mesh-free topology optimization, we use multilayer perceptrons to parameterize both density and displacement fields. Our experiments indicate that our method is highly competitive for minimizing structural compliance objectives, and it enables self-supervised learning of continuous solution spaces for topology optimization problems.

## 1 Introduction

Deep neural networks are starting to show their potential for solving partial differential equations (PDEs) in a variety of problem domains, including turbulent flow, heat transfer, elastodynamics, and many more [1, 2, 3, 4, 5]. Thanks to their smooth and analytically-differentiable nature, implicit neural representations with periodic activation functions are emerging as a particularly attractive and powerful option in this context [4]. In this work, we explore the potential of implicit neural representations for structural topology optimization—a challenging inverse elasticity problem with widespread application in many fields of engineering [6].

Topology optimization (TO) methods seek to find designs for physical structures that are as stiff as possible (i.e. least compliant) with respect to known boundary conditions and loading forces while adhering to a given material budget. While TO with mesh-based finite element analysis is a well-studied problem [7], we argue that mesh-free methods provide unique opportunities for machine learning. We propose the first self-supervised, fully mesh-free method based on implicit neural representations for topology optimization. The core of our approach is formed by two neural networks: a displacement network representing force-equilibrium configurations that solve the forward problem, and a density network that learns optimal material distributions in the domain of interests. To leverage the power of these representations, we cast TO as a stochastic optimization problem using Monte Carlo sampling. Compared to conventional mesh-based TO, this setting

35th Conference on Neural Information Processing Systems (NeurIPS 2021).

introduces new challenges that we must address. To account for the nonlinear nature of implicit neural representations, we introduce a convex density-space objective that guides the neural network towards desirable solutions. We furthermore introduce several concepts from FEM-based topology optimization methods into our learning-based Monte Carlo setting to stabilize the training process and to avoid poor local minima.

We evaluate our method on a set of standard TO problems in two and three dimensions. Our results indicate that neural topology optimization with implicit representations is able to match the performance of state-of-the-art mesh-based solvers. To further explore the potential advantages of this approach over conventional methods, we show how our formulation enables self-supervised learning of continuous solution spaces for this challenging class of problems.

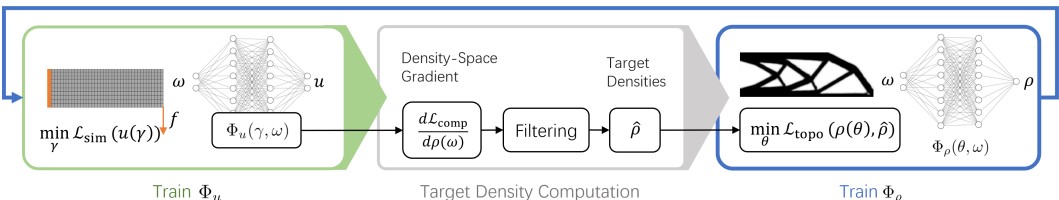

Figure 1: Neural topology optimization pipeline. We compute optimal material distributions by alternately training two neural networks: the displacement network $\Phi_u$ and the density network $\Phi_\rho$, mapping spatial coordinates $\omega$ to equilibrium displacements $u$ and optimal densities $\rho$, respectively. In each iteration, we first update $\Phi_u$ by minimizing the total potential energy of the system. We then perform sensitivity analysis to compute density-space gradients which, after applying our sensitivity filtering, give rise to target density fields $\hat\rho$. Finally, we update $\Phi_\rho$ by minimizing the convex objective $\mathcal{L}_{\text{topo}}$ based on mean squared error between current and target densities.

## 2  Related work

**Neural Networks for Solving PDEs**   Deep neural networks have been widely used in different fields to provide solutions for partial differential equations for both forward simulation and inverse design problems [1, 4, 8]. In this context, PDEs can be solved either in their strong form [9, 10, 11] or variational form [12, 13]. We refer to DeepXDE [5] for a detailed review. Explorations into using deep learning alongside conventional solvers for simulation have been conducted with the goal of accelerating computations [14] or learning the governing physics [15, 16, 17, 18, 19, 20]. With their continuous and analytically-differentiable solution fields, neural implicit representations with periodic activation functions [4] offer a promising alternative to mesh-based finite element analysis. We leverage this new representation to solve high-dimensional inverse elasticity problems in a fully mesh-free manner.

**Differentiable Simulation for Machine Learning**   There is growing interest in differentiable simulation methods that enable physics-based supervision in learning frameworks [21, 22, 23, 24, 25, 15]. Liang *et al*. [22] proposed a differentiable cloth simulation method for optimizing material properties, external forces, and control parameters. Hu *et al*. [21] targets reinforcement learning problems with applications in soft robotics. Geilinger *et al*. [24] proposed an analytically differentiable simulation framework that handles frictional contacts for both rigid and deformable bodies. To reduce numerical errors in a traditional solver Um *et al*. [25] leverage a differentiable fluid simulator inside the training loop. Similar to these existing methods, our approach relies on differentiable simulation at its core, but targets mesh-free, stochastic integration for elasticity problems.

**Neural Representations**   Using implicit neural representation for complex signals has been an ongoing topic of research in computer vision [26, 27], computer graphics [28, 29], and engineering [4, 1]. PIFu [26] learns implicit representations of human bodies for monocular shape estimation. Sitzmann *et al*. [4] use MLPs with sinusoidal activation functions to represent signed distance fields in a fully-differentiable manner. Takikawa *et al*. [30] introduced an efficient neural representation that enables real-time rendering of high-quality surface reconstructions. Mildenhall *et al*. [28] describe a neural radiance field representation for novel view synthesis. We leverage the advantages of implicit

neural representations, demonstrated in these previous works, to learn the high-dimensional solutions of topology optimization problems.

**Topology Optimization with Deep Learning**   Topology optimization methods aim to find an optimal material distribution of a design domain given a material budget, boundary conditions, and applied forces. Building on a large body of finite-element based methods [31, 6, 32, 33, 34, 35, 36, 37, 38, 39, 40, 41, 42, 43, 44], recent efforts have explored the use of deep learning techniques in this context. One line of work leverages mesh-based simulation data and convolutional neural networks (CNNs) to accelerate the optimization process [45, 46, 47, 48, 49, 50, 51]. Perhaps closest to our work is the method by Hoyer *et al*. [52], who reparameterize design variables with CNNs, but use mesh-based finite element analysis for simulation. While Hoyer *et al*. [52] map latent vectors to discrete grid densities, Chandrasekhar *et al*. [53] use multilayer perceptrons to learn a continuous mapping from spatial locations to density values. However, as we show in our comparisons, their choice of using ReLU activation functions leads to overly simplified solutions whose structural performance is not on par with results from conventional methods [54, 55, 56]. In addition, both methods use an explicit mesh for forward simulation and sensitivity analysis, whereas our method is entirely based on neural implicit representations.

## 3   Problem Statement and Overview

Given applied forces, boundary conditions, and a target material volume ratio $\hat{V}$, the goal of topology optimization is to find the material distribution that leads to the stiffest structure. This task can be formulated as a constrained bilevel minimization problem,

$$\mathcal{L}_{\text{comp}}(\rho) = \int_{\Omega} e(\rho, u, \omega) \, d\omega$$
$$\text{s. t. } u(\rho) = \arg\min_{u'} \mathcal{L}_{\text{sim}}(u', \rho), \qquad \rho(\omega) \in \{0, 1\}, \qquad \frac{1}{|\Omega|} \int_{\Omega} \rho \, d\omega = \hat{V} \,, \tag{1}$$

where the loss $\mathcal{L}_{\text{comp}}$ measures how compliant the material is, $\rho$ is the material density field, $\omega$ runs over the domain $\Omega$ and $|\Omega|$ is the volume of $\Omega$. The displacement $u$ is a result of minimizing the simulation loss $\mathcal{L}_{\text{sim}}$, ensuring the configuration is in force equilibrium. $e$ is the pointwise compliance, which is equivalent to the internal energy up to a constant factor and is measuring how much the material is deformed under load; see Section 4.1 for details. Although manufacturing typically demands binary material distributions, densities are often allowed to take on continuous values while convergence to binary solutions is encouraged. We follow the same strategy and parameterize densities $\rho$ and displacements $u$ using implicit neural representations, $\Phi_\rho(\omega; \theta)$ and $\Phi_u(\omega; \gamma)$, respectively. By sampling a batch of locations $\omega^b$ and $\rho^b = \rho(\omega^b)$, we compute an estimate of $\mathcal{L}_{\text{comp}}$ using Monte Carlo integration, $\mathcal{L}_{\text{comp}} \approx \frac{|\Omega|}{n} \sum_i^n e(\omega_i^b)$ . If $u$ is a displacement in force equilibrium, we can compute the total gradient of the compliance loss with respect to the densities of the batch as

$$s = \frac{d\mathcal{L}_{\text{comp}}}{d\rho} = \frac{\partial \mathcal{L}_{\text{comp}}}{\partial \rho} + \frac{\partial \mathcal{L}_{\text{comp}}}{\partial u} \frac{du}{d\rho} = -\frac{\partial \mathcal{L}_{\text{comp}}}{\partial \rho} \,. \tag{2}$$

We will refer to this expression as the *density-space gradient*; see the supplemental document for a detailed derivation. The density-space gradient indicates how the compliance loss changes w.r.t. the density values, assuming that the force equilibrium constraints remain satisfied.

On this basis, we compute the total gradient of the compliance loss with respect to the neural network parameters as $\frac{d\mathcal{L}_{\text{comp}}}{d\theta} = \frac{d\mathcal{L}_{\text{comp}}}{d\rho} \frac{\partial \rho}{\partial \theta}$.

Using this gradient together with a penalty on the volume constraint would be one potential option for solving Equation (1). In practice, however, we observed that this approach does not lead to satisfying behavior. We elaborate on this problem below.

TO is a non-convex optimization problem, whose solutions depend on the optimization method [52] and the path density values take. Much research has been done on developing optimization strategies that converge to *good* local minima for FEM-based solvers. The density-space gradient is generally considered a good update direction for mesh-based approaches. However, when using

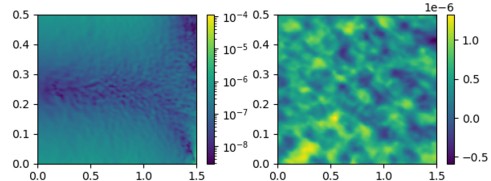

Figure 2: Failure of naive gradient descent. *Left*: negative density-space gradient on the domain for the beam example in the first iteration. Log-scale coloring is used to emphasize structure. *Right*: after one step of gradient descent (learning rate $10^{-5}$), the neural network output has lost all structure from the density-space gradient.

standard optimizers to update the parameters for our density network, the resulting change in densities is not at all aligned with the density-space gradient; see Figure 2. We tested both ADAM [57] and SGD with little difference in the results—eventually, both approaches converge to local minima that are meaningless from a structural point of view; see Section 5.4.

To analyze this unexpected behavior, we consider the change in densities induced by a step in the negative direction of the density-space gradient $\Delta\rho = -\alpha\frac{d\mathcal{L}_{\text{comp}}}{d\rho}$, where $\alpha$ is the learning rate. The corresponding first-order change in network parameters $\Delta\theta$ is $-\alpha\frac{d\mathcal{L}_{\text{comp}}}{d\rho}\frac{\partial\rho}{\partial\theta}$. However, using this parameter change in the Taylor series expansion of the network output, we obtain

$$\Delta\rho' = \rho(\theta + \Delta\theta) - \rho(\theta) = -\alpha\frac{d\mathcal{L}_{\text{comp}}}{d\rho}\frac{\partial\rho}{\partial\theta}\frac{\partial\rho}{\partial\theta}^T + O(\alpha^2) \ . \tag{3}$$

It is evident that, unless the Jacobian $\partial\rho/\partial\theta$ of the neural network is a unitary matrix, the direction of density change $\Delta\rho'$ is different from $\Delta\rho$ even as $\alpha \to 0$.

To avoid converging to bad local minima, we seek to update the network parameters such that the network output changes along the density-space gradient. To this end, we define point-wise density targets $\hat{\rho}^b$ indicating how the network output should change. We then minimize the convex loss function

$$\mathcal{L}_{\text{topo}}(\theta) = \frac{1}{n_b n}\sum_b^{n_b}\sum_j^n \left\|\rho(\omega_j^b;\theta) - \hat{\rho}(\omega_j^b)\right\|^2 \ . \tag{4}$$

While our density-space optimization strategy greatly improves results, we have still observed convergence to minima with undesirable artefacts. Drawing inspiration from mesh-based TO methods [58], we solve this problem with a sensitivity filtering approach (termed $\mathcal{FT}$ in Algorithm 1). To further accelerate convergence, we also adapt the optimality criterion method [7] to our setting ($\mathcal{OC}$ in Algorithm 1). Our resulting neural topology optimization algorithm, which we dubbed *NTopo*, is summarized in Algorithm 1 and further explained in the following Section.

---

**Algorithm 1** NTopo: Neural topology optimization.

---

1: Initialize $\gamma^{(-1)}$, $\theta^{(0)}$ for $\Phi_\rho$, $\Phi_u$; Initialize two optimizers: $\text{opt}_\theta, \text{opt}_\gamma$
2: Run initial simulation: $\gamma^{(0)} \leftarrow \arg\min_\gamma \mathcal{L}_{\text{sim}}(\gamma, \theta^{(0)})$
3: **for** $n_{\text{opt}}$ iterations **do**
4:     **for** $n_{\text{sim}}$ iterations **do** $\gamma^{(l+1)} \leftarrow \text{opt}_\gamma.\text{step}(\gamma^{(l)}, \partial\mathcal{L}_{\text{sim}}/\partial\gamma)$ **end for**
5:     **for** batch $b = 1, .., n_b$ **do** compute $\rho^b$, $s^b$ **end for**
6:     $\hat{s} \leftarrow \mathcal{FT}(\rho^{1,...,n_b}, s^{1,...,n_b})$
7:     $\hat{\rho} \leftarrow \mathcal{OC}(\rho^{1,...,n_b}, \hat{s}^{1,...,n_b}, \hat{V})$
8:     **for** batch $b = 1, .., n_b$ **do** $\theta^{(l+1)} \leftarrow \text{opt}_\theta.\text{step}(\theta^{(l)}, \hat{\rho}^b, \partial\mathcal{L}_{\text{topo}}/\partial\theta)$ **end for**
9: **end for**
10: **return** $\Phi_\rho$

---

## 4 Neural Topology Optimization

We start the technical description of our algorithm with a brief overview of the neural representation that we use. We then explain how to compute equilibrium configurations and how to update the density field. Finally, we introduce an extension of our method from individual solutions to entire solution spaces for a given continuous parameter range.

We use SIREN [4] as neural representation, which is a dense multilayer perceptron (MLP) with sinusoidal activation functions. A SIREN-MLP with $l$ layers, $l-1$ hidden layers, and $h$ neurons in each hidden layer is defined as

$$\Phi(x) = W_l(\phi_{l-1} \circ \phi_{l-2} \circ \ldots \circ \phi_0)(\omega_0 x) + b_l, \qquad \phi_i(x) = \sin(W_i x_i + b_i) \tag{5}$$

where $x$ is the input vector, $y$ is the output vector, $W_i$ are weight matrices, $b_i$ are biases, and $\omega_0$ is a frequency dependent factor. We use a standard five-layer MLP with residual links and no regularization. The weights of all layers are initialized as suggested by Sitzmann *et al.* [4].

## 4.1 Computing Static Equilibrium Solutions

We parametrize the displacement field $u$ using a neural network $\Phi_u(\omega; \gamma)$ with network weights $\gamma$. To find the displacement field $u$ in static equilibrium, we minimize the variational form of linear elasticity which is given by

$$\min_\gamma \mathcal{L}_{\text{sim}}(u(\gamma)) = \min_\gamma \int_\Omega \frac{1}{2} \varepsilon(u) : \sigma(u) - u^T f \, d\omega \qquad \text{s.t.} \quad u|_{\partial\Omega_D} = u_D \tag{6}$$

where $\partial\Omega_D$ is the part of the boundary of the domain with prescribed displacement $u_D$. The internal energy in linear elasticity is given through $\frac{1}{2}\varepsilon : \sigma$, where the tensor contraction ":" is defined as $\epsilon : \sigma = \sum_{ij} \epsilon_{ij}\sigma_{ij}$. In two dimensions we compute the stress tensor under plane stress assumption $\sigma = \left(E/\left(1-\nu^2\right)\right)((1-\nu)\varepsilon + \nu \operatorname{trace}(\varepsilon)\mathbf{I})$ where $\nu$ is Poisson's ratio, $E$ is the Young's modulus, $\mathbf{I} \in \mathbb{R}^{2\times2}$ is the identity matrix and $\varepsilon = \left(\nabla u + \nabla u^T\right)/2$ is the linear Cauchy strain. In 3D we use Hooke's law $\sigma = \lambda \operatorname{trace}(\varepsilon)\mathbf{I} + 2\mu\epsilon$ where $\lambda = E\nu/((1+\nu)(1-2\nu))$ and $\mu = E/(2(1+\nu))$. Following SIMP [37], we parameterize the Young's modulus $E$ using the density field as $E(\rho) = \rho^p E_1$. Larger values for the exponent $p$ together with the volume constraint encourage binary solutions.

**Sampling** To evaluate the integral in Equation (6) we resort to a Quasi-Monte Carlo sampling approach. We generate stratified samples on a grid with $n_x \times n_y$ cells in 2D (and $n_x \times n_y \times n_z$ in 3D). We adjust $n_x, n_y, n_z$ to match the aspect ratio of the domain.

**Enforcing Dirichlet Boundary Conditions** By constructing a function $d$ that is zero on the fixed boundary and an interpolator of the function $u_D$, we enforce the displacement field $u(\omega) = d(\omega)\Phi_u(\omega; \gamma) + (I \circ u_D)(\omega)$ to always satisfy the essential boundary conditions, thus turning the constrained problem into an unconstrained one [59]. We use simple boundaries in our examples for which analytic functions $d$ are readily available; see the supplemental document for more details.

## 4.2 Density Field Optimization

We reparameterize the density field using a neural network $\Phi_\rho$ which maps spatial locations to their corresponding density values. The bound constraints on the densities are enforced by applying a scaled logistic sigmoid function to the network output, specifically $\rho(\omega) = \operatorname{sigmoid}(5\,\Phi_\rho(\omega))$. The total volume constraint is satisfied by the optimality criterion method described below.

**Moving Mean Squared Error (MMSE).** Equation (4) can be interpreted as a mean squared error

$$\frac{1}{|\Omega|} \int_\Omega ||\rho(\omega; \theta) - \hat{\rho}(\omega)||^2 \, d\omega \, . \tag{7}$$

We minimize this loss using a mini-batch gradient descent strategy, where we use every batch only once. We collect multiple batches of data from $\hat{\rho}$ before we update $\rho$ and $\hat{\rho}$, specifically $\hat{\rho}$ changes once every outer iteration. For this reason, we refer to this updating scheme as moving mean squared error in the following.

**Sensitivity filtering $\mathcal{FT}$.** In conventional TO algorithms, filtering is an essential component for discovering desirable minima that avoid artefacts such as checkerboarding [33]. While our neural representation does not suffer from the same discretization limitations that give rise to checkerboard patterns, we have nevertheless observed convergence to undesirable minima. Indeed, the neural

representation alone does not remove the inherent reason for such artefacts: TO is an underconstrained optimization problem with a high-dimensional null-space. Isolating good solutions from this null-space requires additional priors, filters, or other regularizing information.

In order to address this problem, we propose a *continuous* sensitivity filtering approach that, instead of using discrete approximations [58], is based on continuous convolutions. Following this strategy, we obtain filtered sensitivities as

$$\hat{s}(\omega) = \frac{\int H(\Delta\omega)\rho(\omega + \Delta\omega)s(\omega + \Delta\omega)\,d\Delta\omega}{\max(\epsilon, \rho(\omega)) \int H(\Delta\omega)\,d\Delta\omega} \tag{8}$$

where $\epsilon$ is set to $10^{-3}$ and $H$ is the kernel $H(\Delta\omega) = \max(0, r - \|\Delta\omega\|)$, with radius $r$. Since the samples $\omega^i$ are distributed inside a grid, we can compute an approximation to the continuous filter as

$$\hat{s}_j^i = \frac{\sum_{k\in N^j} H(\omega_j^i - \omega_k^i)\rho_k^i s_k^i}{\max(\epsilon, \rho_j^i) \sum_{k\in N^j} H(\omega_j^i - \omega_k^i)} \tag{9}$$

where the neighborhood $N$ is defined by cell sizes and radius $r$ such that points inside the footprint of the kernel $H$ are in the neighborhood $N$. Although this approximation is not an unbiased estimator of Equation (8), it led to satisfying results in all our experiments.

**Multi Batch-based Optimality Criterion Method $\mathcal{OC}$**  We leverage the optimality criterion method [36] to compute density targets that automatically satisfy volume constraints, thus avoiding penalty functions or other constraint enforcement mechanisms. To this end, we extend the discrete, mesh-based formulation to the continuous Monte Carlo setting. One chooses a Lagrange multiplier $\lambda$ such that the the volume constraint is satisfied after the variables have been updated. Since it is computationally infeasible to compute the constraint exactly, we choose to satisfy the constraint in terms of its estimator using the collected batches. Additionally, we adopt a heuristic updating scheme very similar to the ones proposed by other authors [36], which leads to the following scheme: First a set of multiplicative updates $B^i$ are computed, which then are applied to compute the target densities $\hat{\rho}^i = \mathrm{clamp}(\rho^i(B^i)^\eta, \max(0, \rho^i - m), \min(1, \rho^i + m))$, where $m$ limits the maximum movement of a specific target density and $\eta$ is a damping parameter. We used $m = 0.2$ and $\eta = 0.5$. The updated $B^i$ are computed using $B^i = -\hat{s}^i/(\lambda\frac{\partial V}{\partial\rho^i})$ where $\lambda$ is found using a binary search such that the estimated volume of the updated densities using Monte Carlo integration matches the desired volume ratio across all batches $\frac{1}{n_b n}\sum_b^{n_b}\sum_j^n \hat{\rho}_j^b = \hat{V}$.

### 4.3  Continuous Solution Space

Apart from solving individual TO problems for fixed boundary conditions and material budgets, our method can be readily extended to learn entire spaces of optimal solutions, *e.g.*, a continuous range of material volume constraints $\{\hat{V}^i\}$ or boundary conditions such as force locations. To this end, we seek to find a density function $\rho(\omega, q)$ which yields the optimal density at any point $\omega$ in the domain for any parameter vector $q$ representing, e.g., material volume ratio $q := \hat{V}$ in the target range $Q = [\underline{V}, \overline{V}]$. In a supervised setting, a common approach is to first compute $k$ solutions corresponding to different parameters $\{q^k\}$ and then fit the neural network using a mean squared error. By contrast, our formulation invites a fully self-supervised approach based on a modified moving mean squared error,

$$\frac{1}{|Q||\Omega|} \int_Q \int_\Omega \|\rho(\omega, q; \theta) - \hat{\rho}(\omega, q)\|^2\, d\omega dq\,. \tag{10}$$

We minimize this loss by sampling $q$ at random and then update the density network using the same method as described for the single target volume case.

## 5  Results

To analyze our method and evaluate its results, we start by comparing material distributions obtained with our approach on a set of 2D examples. We demonstrate the effectiveness of our method through comparisons to a state-of-the-art FEM solver (Section 5.1). We then investigate the ability of our formulation to learn continuous spaces of solutions in a fully self-supervised manner. Comparisons

to a data-driven, supervised approach indicate better efficiency for our method, suggesting that our approach opens new opportunities for design exploration in engineering applications (Section 5.2). We then turn to TO problems with non-trivial boundary conditions and demonstrate generalization to 3D examples (Section 5.3). An ablation study justifies our choices of using sensitivity filtering and casting the nonlinear topology objective into an MMSE form (Section 5.4). We further compare the impact of different activation functions and provide detailed descriptions of our learning settings (Section 5.5).

## 5.1 Comparisons with FEM Solutions

We demonstrate that our results are competitive to those produced by a SIMP, a reference FEM approach [36] for mesh-based topology optimization. As can be seen in Figure 3, results are qualitatively similar, but our method often finds more complex supporting structures that lead to lower compliance values (see Table 1). To put these results in perspective, there is no topology optimization method fundamentally better than SIMP, despite decades of research. For fair comparisons, the compliance values of these structures are evaluated using the FEM solver and we consistently use fewer degrees of freedom (DoFs). The DoFs in these two methods are the number of network weights and the number of finite element cells, respectively. As can be seen, our method is more computationally expensive, but we would like to emphasize that the goal of our approach is not to outperform conventional solvers for single-solution cases, but rather find insight for a learning-based and fully mesh-free approach to topology optimization that allows for self-supervised learning of solution spaces. We refer to the supplemental material for further details of these examples.

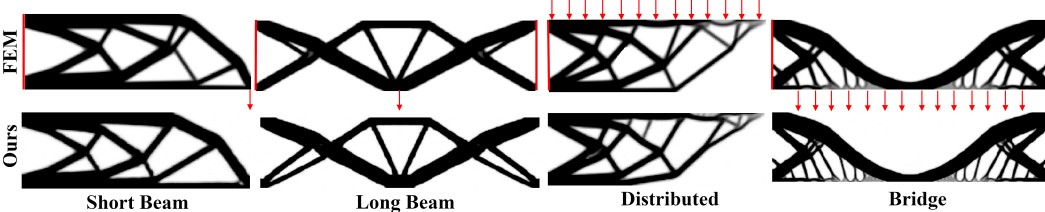

Figure 3: 2D comparison on four example problems. Solutions produced by our method are qualitatively equivalent to corresponding FEM solutions. Force and Dirichlet boundary conditions are visualized in the top-row images.

Table 1: Statistics of 2D comparisons. Our results achieve quantitatively lower compliance values (Comp.) than the reference solver (SIMP) for all examples.

| | FEM | | | | OURS | | | |
|---|---|---|---|---|---|---|---|---|
| EXP. | COMP. | DOFS | ITER. | TIME | COMP. | DOFS | ITER. | TIME |
| SHORTBEAM | $1.173 \times 10^{-3}$ | 30,000 | 122 | 34 s | $1.166 \times 10^{-3}$ | 28,300 | 200 | 33 min |
| LONGBEAM | $2.595 \times 10^{-4}$ | 31,250 | 155 | 47 s | $2.592 \times 10^{-4}$ | 28,300 | 200 | 33 min |
| DISTRIBUTED | $2.026 \times 10^{1}$ | 30,000 | 454 | 114 s | $2.012 \times 10^{1}$ | 28,300 | 400 | 66 min |
| BRIDGE | $4.441 \times 10^{0}$ | 31,250 | 233 | 72 s | $4.385 \times 10^{0}$ | 28,300 | 200 | 33 min |

## 5.2 Learning Continuous Solution Spaces

**Optimal designs for different volume constraints** Here we demonstrate the capability of our method to learn continuous spaces of optimal density distributions for a continuous range of material volume constraints. We minimize the objective defined in Equation (10) for volume constraint samples drawn randomly from the range $[30\%, 70\%]$. To evaluate the accuracy of our learned solutions, we apply our single-volume algorithm for 11 discrete material volume constraints sampled uniformly across the target range. As can be seen in Figure 5, our solution space network does not compromise the quality of individual designs. The mean and maximum errors in compliance and volume violation are $0.75\%$, $3.83\%$, $0.5\%$ and $2.83\%$, respectively. We therefore conclude that our model successfully learned the continuous solution space. Furthermore, we argue that the level of accuracy is acceptable for design exploration in many engineering applications.

**Different solution spaces** To further analyze the behavior of our solution space approach, we conducted two additional experiments: the beam example with fixed volume but varying location for the applied forces and the bridge example with varying density constraint; see Figure 4. Both examples confirmed our initial observations, showing smoothly evolving topology and compliance values close to the single-solution reference. For both examples, we sample 25 volume fractions/force locations during each iteration using stratified sampling. In addition to our single solution setup, we shuffle the density pairs randomly during the MMSE fitting step. The training takes 280 iterations in total, leading to 37.3 hours. Once trained, the inference enables real-time exploration of the solution space (0.014s/71.4FPS for $300 \times 100$ samples).

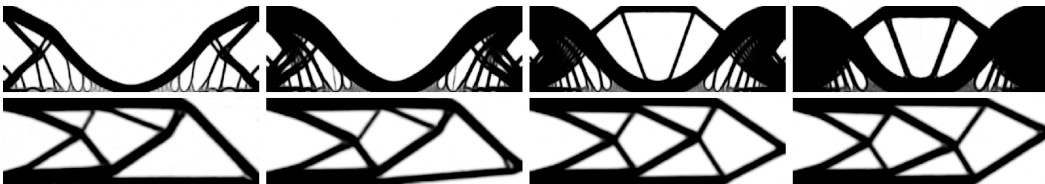

Figure 4: Different solution spaces. Here we show additional results varying the volume fraction constraints for the bridge example (first row) as well as varying the applied force location for the cantilever beam (second row). See Figure 3 for boundary conditions.

**Comparison with supervised setting** We further compare our self-supervised training techniques with a supervised setting under the same computational budget. The cost of performing $1,000$ optimization steps ($5.25h$) with our larger solution space network is similar to (but somewhat lower than) the cost of computing 11 solutions under single volume constraints. We select 11 volume constraint values uniformly and train the network from these solutions in a supervised fashion. As can be seen in Figure 6, the network performs poorly for unseen data leading to infeasible designs and significant volume violations. On the contrary, our self-supervised approach leads to physically valid material distributions.

## 5.3 Irregular BCs and 3D Results

Our formulation extends to more complex boundaries, which we illustrate on a set of additional examples. Figure 8 shows multiple examples where densities are constrained on circu-

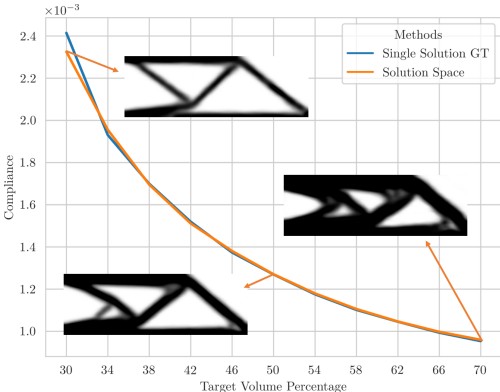

Figure 5: Results and compliance comparisons between evaluating our solution space network at discrete volume locations and the reference solution produced by our method in the single volume constraint case. As can be seen, the learned solution space does not compromise the quality of individual solutions.

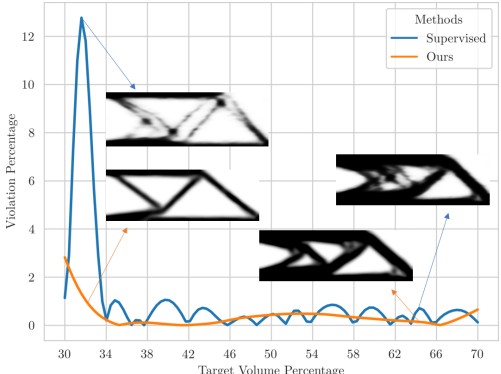

Figure 6: Comparisons with supervised setting. Our self-supervised learning methods outperforms its supervised counterparts with less computational cost. The target volume constraint is shown on the $x$-axis and the constraint violation (in percentage) of the resulting structures is given on the $y$-axis.

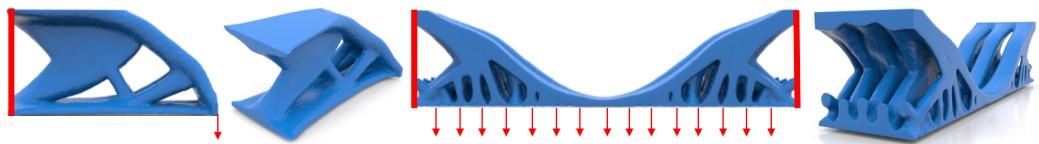

Figure 7: The solutions for two 3D examples demonstrate the promise of our method in 3D. A 3D cantilever beam (left) and a 3D bridge (right). The mesh has been generated using a marching-cubes algorithm [60].

lar sub-domains and the Dirichlet boundary $\partial\Omega_D$ is also circular in one of the two examples. Our method shows promise in 3D, as demonstrated on the two examples shown in Figure 7. Due to the symmetry of the configuration, we apply symmetric boundary conditions to reduce the domain of interests to half and quarter for the cantilever beam and bridge example respectively to save computational cost and memory usage. Our method finds smooth solutions with various supporting features for the two tasks.

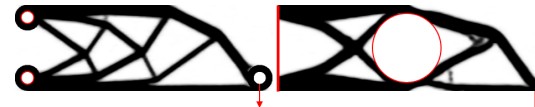

Figure 8: Curved boundaries: In the left example 3 holes have been put in the design using constraints on the density field. On the right, the density field is constrained to have a hole in the middle.

### 5.4 Ablation Studies

Here we provide evidence for the necessity of using a sensitivity filter, our moving mean squared error loss during the optimization process, and the influence of different neural networks. In the first test, we do not rely on the optimality criterion method to compute the target density field nor do we use the mean squared error. Rather, we add a soft penalty term to satisfy the volume constraint and update the neural network only once per iteration. In the second test, we adopt the proposed method without filtering. Comparing with our reference solutions at different iterations, the alternatives either converge significantly slower or arrive at poor local minima with many artefacts, such as disconnected struts or rough boundaries. We further compare our network structure with a ReLU-MLP as proposed in TOuNN [53]. As can be seen from Figure 10, this approach fails to capture much of the geometric features that our method (and SIMP) produce, leading to more compliant (i.e., less optimal) designs (compliance: 0.001196). The Fourier feature network [29] leads to comparable results (compliance: 0.001172) and is thus also a valid option for our method.

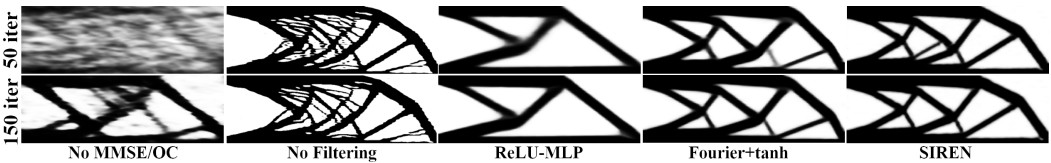

Figure 9: Ablation studies on the beam example. $1st$ column: without moving mean squared error and optimality criterion—the density-space gradient is directly applied to the network, which is updated once per optimization iteration. $2nd$ column: proposed method without filtering. $3rd$ column: proposed method using ReLU-MLP as the density network, SIREN is adopted as simulation network to obtain accurate equilibrium displacement. $4th$ column: Proposed method with Fourier feature network. $5th$ column: Proposed method with SIREN network. We verify that our moving mean squared error, filtering, and choice of activation function are all quintessential.

### 5.5 Training Details.

We use Adam [57] as our optimizer for both displacement and density networks and the learning rate of both is set to be $3 \cdot 10^{-4}$ for all experiments. We use $\omega_0 = 60$ for the first layer and 60 neurons in each hidden layer in 2D, and 180 hidden neurons in 3D. For the solution space learning setup, we use 256 neurons in each hidden layer in the density network to represent the larger solution space. For all experiments, we initialize the output of the density network close to a uniform density distribution of the target volume constraint by initializing the weights of the last layer close to zero and adjusting

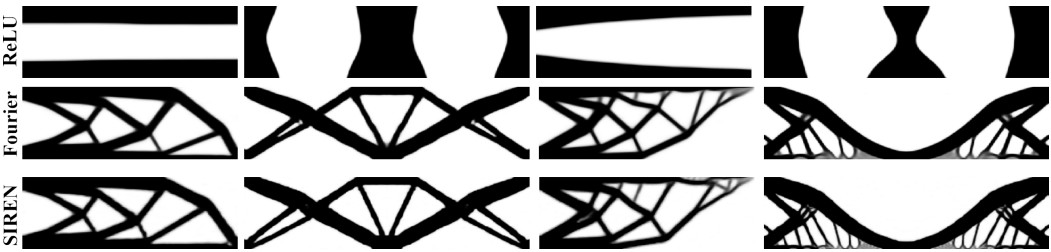

Figure 10: We use different network architectures to parameterize both the density and displacement field. Since the ReLU-MLP fails to capture the high-frequency details, its solution for the forward problem is far from being accurate, resulting in meaningless structures for the inverse problems. Fourier feature and SIREN networks produced similar results, thus, suitable for our method.

the bias accordingly. We used $E_1 = 1, \nu = 0.3, p = 3, n_b = 50$ and $[n_x, n_y] = [150, 50]$ in 2D and $[n_x, n_y, n_z] = [80, 40, 20]$ in 3D. The training iterations for the 3D examples are 100 with a per iteration cost of $131.7s$. All timings in the paper are reported on a GeForce RTX 2060 SUPER graphics card.

## 6 Conclusion

We propose a novel, fully mesh-free TO method using implicit neural representations. At the heart of our method lie two neural networks that represent force-equilibrium configurations and optimal material distribution, respectively. Experiments demonstrate that our solutions are qualitatively on par with the standard FEM method for structural compliance minimization problems, yet further enables self-supervised learning of solution spaces. The proposed method can handle irregular boundary conditions due to its mesh-free nature and is applicable to 3D problems as well. As such we consider it a steppingstone towards solving many varieties of inverse design problems using neural networks.

**Limitations and Future Works**   As we adopted the sigmoid function to enforce box constraints, it naturally leads to small gradients when approaching 0 or 1. Although it did not lead to convergence issue for us in practice, better ways of enforcing box constraints in density space is an interesting avenue for future work. Like for other Monte Carlo-based methods, advanced sampling strategies, *e.g.*, importance sampling can also be explored to speed up the optimization process.

## Acknowledgements

This research was supported by the Discovery Accelerator Awards program of the Natural Sciences and Engineering Research Council of Canada (NSERC), the European Research Council (ERC) under the European Union's Horizon 2020 research and innovation program (Grant No. 866480) and the Swiss National Science Foundation (Grant No. 200021_200644).

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
