# OpenReview forum: "NTopo: Mesh-free Topology Optimization using Implicit Neural Representations"
_NeurIPS.cc/2021/Conference — NeurIPS 2021 Poster_

### Official Review · Reviewer_F4BW · 2021-07-16

**Rating:** 6
**Confidence:** 3

**Summary:**

This work introduces a method for topology optimization based on implicit neural representations.
The goal of topology optimization generally is to find optimal material distribution that produces the structure which is as stiff as possible. The key idea presented in this work is to use two neural networks: the density network, which implicitly represents optimal material density field, and the displacement network, which implicitly represents displacement field.
The optimization algorithm then proceeds by alternatively training these two networks, while applying additional filtering and ad-hoc supervision to steer the process to better optima. Evaluation is conducted on a set of standard 2D topology optimization tasks.

**Limitations And Societal Impact:**

Authors did discuss minor limitations of their approach.

**Main Review:**


### Paper writing / clarity
**-** Paper is often hard to follow, especially for people who are not closely familiar with topology optimization. See the list of minor issues below which could help improve readability / clarity.

### Motivation / novelty
**+** Using neural nets to approximate solutions to complex physical systems is in general a very promising direction, and it seems like implicit neural representations quite naturally fit the topology optimization domain.

**-** However, it appears that the idea of using a neural network to implicitly model optimal density field and use those in an optimization loop has been explored before [53]. It is not clear if simply using SIREN in place of a relu MLP is a sufficiently large technical contribution, even if it does improve performance. It would be great if authors discuss other key differences between their proposed framework and "TOuNN".

### Evaluation
**+** Numerical evaluation indicates that proposed method outperforms a strong baseline - SIMP by a small margin. However, the resulting solutions seem to be _very_ similar qualitatively to the baseline. It would be great to have some context for non-domain experts to understand how large an improvement is reported in Table 1.

**-** Discussion of efficiency of the proposed approach with respect to the baseline (SIMP) is not present.

**-** Evaluation in 3D settings is not very convincing: authors report an example without any numerical comparisons to any baselines. At very least this should be addressed by changing the claims.

**+** I liked the examples of continuously changing volume constraints in the supplementary: if those can be obtained in real-time, it might make sense to elaborate on this in the main paper (and also would be great to know if competing approaches such as [53] and SIMP could get to similar results).

### Minor
* Might be a good idea to adapt notation a little bit to indicate which quantities are vectors vs scalars (e.g. use bold).
* L29: domain of interest
* L78: "use fully-connected layers": you mean MLPs?
* L90: notation ":" confused me until I reached 4.1.
* Notations in "Algorithm 1" are a bit too dense and confusing. Maybe makes sense to unroll the "opt" notation for better readability.
* L146-147: what is meant by "standard MLP with residual links"? if my understanding is correct, SIRENs do not use residual connections.
* L170: why 5 ? mb makes sense to introduce a notion for this scalar?


**Time Spent Reviewing:**

3

---

> ### Author Response · Authors · 2021-08-05
> **Response**
>
> **Q1: However, it appears that the idea of using a neural network to implicitly model optimal density field and use those in an optimization loop has been explored before [53]. It is not clear if simply using SIREN in place of a relu MLP is a sufficiently large technical contribution, even if it does improve performance. It would be great if authors discuss other key differences between their proposed framework and "TOuNN".**
>
> A1: To the best of our knowledge, our method is the first topology optimization approach to use neural networks for representing (and optimizing) both density fields and displacement. By combining neural implicit representations with a Monte-Carlo formulation for simulation and optimization, our method is fully mesh-free - a fundamental difference to TOuNN, which requires a finite element mesh to perform simulation and sensitivity analysis. We further demonstrate that simply substituting the mesh-based FEM solver in TOuNN with the same ReLU-MLP that is used for densities is problematic and leads to a severe loss of details in the resulting structures. We overcome these non-trivial technical challenges with our method.
>
> **Q2: Discussion of efficiency of the proposed approach with respect to the baseline (SIMP) is not present.
> Evaluation in 3D settings is not very convincing: authors report an example without any numerical comparisons to any baselines. At very least this should be addressed by changing the claims.**
>
> A2: For timings with SIMP and quantitative comparisons in 3D, please kindly refer to the answers to reviewer YJ3u.
>
> **Q3: Numerical evaluation indicates that proposed method outperforms a strong baseline - SIMP by a small margin. However, the resulting solutions seem to be very similar qualitatively to the baseline. It would be great to have some context for non-domain experts to understand how large an improvement is reported in Table 1.**
>
> A3: We refer to the answer to the reviewer YJ3u for a clarification of the objective of the proposed method.
>
> **Q4: I liked the examples of continuously changing volume constraints in the supplementary: if those can be obtained in real-time, it might make sense to elaborate on this in the main paper (and also would be great to know if competing approaches such as [53] and SIMP could get to similar results).**
>
> A4: As we reported in Line 254 of the main paper, the evaluation of our learned solution spaces is indeed fast enough for real-time applications. Since SIMP is a method for obtaining a single solution, it cannot represent solution spaces by itself. To obtain the material distribution at arbitrary locations, one has to solve a new topology optimization problem each time. Please kindly refer to the answers to reviewer YJ3u to see regarding timings.
>
> **Other:**
>
> + Might be a good idea to adapt notation a little bit to indicate which quantities are vectors vs scalars (e.g. use bold).
>   - We will update it in the final version
> + L29: domain of interest
>   - We will update it in the final version
> + L78: "use fully-connected layers": you mean MLPs?
>   - Yes
> + L90: notation ":" confused me until I reached 4.1.
>   - We included “see Section 4.1 for details” but a better ordering can be arranged for the final version
> + Notations in "Algorithm 1" are a bit too dense and confusing. Maybe makes sense to unroll the "opt" notation for better readability.
>   - We will update it in the final version
> + L146-147: what is meant by "standard MLP with residual links"? if my understanding is correct, SIRENs do not use residual connections.
>   - We found in practice that residual links lead to slightly better performance.

---

> > ### Comment · Reviewer_F4BW · 2021-08-23
> > **re:**
> >
> > Thanks for a detailed response. Most of my concerns have been addressed. I am thus inclined to increase my original rating.

---

### Official Review · Reviewer_jpbD · 2021-07-16

**Rating:** 6
**Confidence:** 2

**Summary:**

The paper proposes a method for topology optimization (TO) in solid mechanics by modeling displacement and material densities as implicit neural networks. TO is a very important field in mechanics and it is exciting to see successful ML applications in the area. The paper shows that their method can solve for topologies under various load and displacement constraints in both 2D and 3D.


**Limitations And Societal Impact:**

Limitations were discussed but potential negative societal impact needs to be added.

**Main Review:**

## Strengths

- The results appear to be very strong on a variety of cases compared to state-of-the-art methods.

## Weaknesses

- The paper seems to be targeted to topology optimization audience and as such lacks background on a lot of concepts. Specifically I think the multi-batch optimality criterion lacks details and should perhaps be alluded to before (4). A figure on how sensitivity filtering helps might make it more intuitive for a reader.

## Questions

- In Appendix 2, how does the Lagrange multiplier in (4) become the strain in (5)? Also is there a reason why not accounting for u's dependence on $\rho$ is okay in (9). Some additional clarity here would be useful as this defines the density space gradients which is key to the paper.

- How sensitive are the methods to initialization of the network?

- The sensitivities are first introduced in Algorithm 1 and then in (8); can you elaborate on what these are?

## Originality

The key idea seems novel and solid.

## Clarity

A lot of the sections in the paper were not clear to me (see weaknesses). I think the paper will do well if written with a broader audience at NeurIPS in mind.

**Time Spent Reviewing:**

10-12 hours

---

> ### Author Response · Authors · 2021-08-05
> **Response**
>
> **Q1: The paper seems to be targeted to topology optimization audience and as such lacks background on a lot of concepts. Specifically I think the multi-batch optimality criterion lacks details and should perhaps be alluded to before (4). A figure on how sensitivity filtering helps might make it more intuitive for a reader.**
>
> A1: We are happy to improve clarity along the lines suggested. The optimality criterion approach is a heuristic density update scheme that is common practice in the topology optimization community. The idea is to efficiently update density values while satisfying the volume constraint without the need for a penalty term. We adopt this concept and adapt it to our learning-based framework. More elaboration can be added in the final version.
>
> **Q2: In Appendix 2, how does the Lagrange multiplier in (4) become the strain in (5)?**
>
> A2: The step from (4) to (5) is purely algebraic and relies on the fact that $\sigma$ is symmetric. It then holds that $\nabla \lambda : \sigma = \epsilon(\lambda) : \sigma$ where $\epsilon$ is the function $\epsilon(\lambda)=\frac{1}{2}\left(\nabla \lambda + \nabla {\lambda}^T \right)$.
>
> **Q3: Also is there a reason why not accounting for u's dependence on ρ is okay in (9). Some additional clarity here would be useful as this defines the density space gradients which is key to the paper.**
>
> A3: The displacement field $u$ depends on the density field $\rho$ through equilibrium conditions. The difference in formulations here is due to the choice of
> 1) using separate variables, $\rho$ and $u$, and introducing explicit equilibrium constraints vs.
> 2) eliminating constraints and keeping just one set of variables ($\rho$).
>
> Version 1), expressed in Eq. (9, Appendix), for which $u$ is independent of $\rho$, corresponds to the standard formulation of constrained optimization problems using Lagrange multipliers. In the corresponding Lagrangian, all arguments ($u$, $\rho$, $\lambda$, $\mu$) are independent and are only coupled  through the solution condition $\nabla \mathcal{L} = 0$. Version 2) corresponds to the constraint elimination approach (using the implicit function theorem), where $u = Simulation(\rho)$ is a function of the densities and there are no constraints. We will clarify the exposition to better distinguish between these two viewpoints on the optimization problem.
>
> **Q4: How sensitive are the methods to initialization of the network?**
>
> A4: The initialization of the SIREN network has been studied by Sitzmann et al., and we followed the strategy suggested in their work. Although we did not formally study the impact of initialization, we have not observed robustness issues or problems related to high sensitivities in our experiments.
>
> **Q5: The sensitivities are first introduced in Algorithm 1 and then in (8); can you elaborate on what these are?**
>
> A5: The loss depends on the equilibrium displacement field, which is implicitly coupled to the density values $\rho(\omega)$ through equilibrium constraints. The sensitivities give us the pointwise gradient of the loss with respect to these density values, accounting for the coupling between displacement field and density values.

---

### Official Review · Reviewer_YJ3u · 2021-07-19

**Rating:** 6
**Confidence:** 3

**Summary:**

In this work, neural networks with sinusoidal activation functions are used as continuous function approximators for both the density and displacement fields in structural topology optimization problems. This allows for the development of a mesh-free topology optimization procedure, which is claimed to be on par with existing traditional FEM-based approaches, while also allowing for the learning of continuous solution spaces over certain parameters.

**Limitations And Societal Impact:**

Societal impact was not discussed and claimed to be N/A.

The work is lacking some discussion of possible limitations, these are discussed in the main comments above (with respect to comparison to other methods). The authors should address these.

**Main Review:**

This work is clearly presented.

The authors method seems well motivated, and draws from recent advances in physics informed neural networks and the utilization of periodic activation functions (SIRENs). However, supposedly given the complexity of the optimization tasks being studied, a number of additional tweaks are necessary to ensure good behavior of the optimization algorithm. These are studied in ablation studies in the experimental section.

The empirical results are the core of the paper. Below I comment on these, and raise some questions which I hope the authors will address.

 First, the authors demonstrate that the proposed method is able to achieve results on par (or slightly better) than traditional FEM-based approaches. These are good results, yet it would be important for the authors to also have made available the comparison along other axes that make the tradeoffs between methods more evident. For example, what are the computational costs and runtime of each, both for training and evaluation (for the machine learning-based approach). Similarly for the "learning continuous solution spaces", how would the run time of a traditional solver compare here? (Maybe for solving each of the evaluated positions). It would be important to have these values so that the reader can have a clearer picture of what are the pros and cons of each method.

The 3D evaluation seems qualitatively good, yet quantitative results are not presented, and a comparison against the traditional methods is also absent. (An analysis of computational cost tradeoffs, as mentioned in the previous paragraph would be interesting in this case too.) Given this, the claim in the conclusion that the "can generalize well to 3D" (line 303) seems a bit strong, given the brevity (and qualitative nature) of the experimental results in this area.

The ablation studies seem to support the claims of the authors. However, to make the results more clearly interpretable, it would probably be a good idea to combine the values reported into lines 283 and 284 into Table 1 (or a similar table), where the reader could easily see the comparison across methods with similar units. Additionally, it is not clear to me why results on these baselines were only reported on one experiment. As mentioned previously, it would probably make sense to have a filled in Table 1 (or similar table) with the values for baselines for all experiments, so that a more comprehensive comparison can be performed, not only between the proposed methods and traditional FEM approaches, but also to the baselines.

Despite the issues raised in the previous paragraphs, I believe the methods presented in this paper are interesting and the results look promising. As such, I am willing to improve my evaluation if the authors address these concerns.

-------------

Update after authors' response:

The authors have addressed the concerns I raised above, and as such I am updating my rating of the paper.

**Time Spent Reviewing:**

I do not review my assigned papers sequentially and thus have not estimated how much time was spent on each paper

---

> ### Author Response · Authors · 2021-08-05
> **Response**
>
> **Q1: These are good results, yet it would be important for the authors to also have made available the comparison along other axes that make the tradeoffs between methods more evident. For example, what are the computational costs and runtime of each, both for training and evaluation (for the machine learning-based approach).**
>
> A1: We are happy to expand the comparison between our method and the conventional FEM-based method SIMP as suggested. The timings and iteration counts for SIMP are
>
> short beam 34s, iteration count 122; long beam 47s, iteration count 155; distributed 114s, iteration count 454;
> bridge 72s iteration count 233.
>
> These timings are reported for an Intel i7-9700K at grid resolutions documented in Table 1.
>
> Our training time and iteration count are given in the main paper (Line 294):
>
> short beam 33m, iteration count 200; long beam 33m, iteration count 200; distributed 66m, iteration count 400;
> bridge 33m iteration count 200.
>
> The evaluation of our learning-based approach takes 0.012s for 300 x 100 samples, both timings are reported using an Nvidia RTX 2060 SUPER.
>
> We would like to emphasize that the goal of our approach is not to outperform conventional solvers for single-solution cases. Rather, the main insights that we hope to share through this work are that
> 1) a learning-based and fully mesh-free approach to topology optimization can deliver results that are on par with the conventional mesh-based methods, and that
> 2) this new approach allows for efficient, self-supervised learning of continuous solution spaces which SIMP cannot.
>
> **Q2: Similarly for the "learning continuous solution spaces", how would the run time of a traditional solver compare here? (Maybe for solving each of the evaluated positions).**
>
> A2: The conventional solver (SIMP) for given parameters takes 34s for the force-varying example of the Short Beam, and 72s for the volume-varying example of the Bridge. With our method, evaluating the learned solution space at arbitrary parameter locations takes 0.014s, thus enabling real-time exploration.
>
> **Q3: The 3D evaluation seems qualitatively good, yet quantitative results are not presented, and a comparison against the traditional methods is also absent. (An analysis of computational cost tradeoffs, as mentioned in the previous paragraph would be interesting in this case too.)**
>
> A3: We can include quantitative evaluations of our 3D results and perform comparisons to the conventional solver for the 3D examples if needed.
>
> **Q4: It would probably be a good idea to combine the values reported into lines 283 and 284 into Table 1 (or a similar table), where the reader could easily see the comparison across methods with similar units.  Additionally, it is not clear to me why results on these baselines were only reported on one experiment. As mentioned previously, it would probably make sense to have a filled in Table 1 (or similar table) with the values for baselines for all experiments, so that a more comprehensive comparison can be performed, not only between the proposed methods and traditional FEM approaches, but also to the baselines.**
>
> A4: We are happy to include quantitative comparisons as suggested. We did not quantitatively compare the proposed method on different network structures on all examples as this ablation study is intended to show that the proposed method is not limited to one particular activation function or network architecture, as both SIREN and Fourier feature networks produce similar results.
> Quantitative comparison against the ReLU-MLP baseline is not possible as it failed to converge to physically meaningful results - kindly refer to Figure 10.

---

> > ### Comment · Reviewer_YJ3u · 2021-08-30
> > **Response**
> >
> > Thanks for the response. These additional details and changes address the concerns I had and so, as I had indicated previously, I will update my evaluation.

---

### Comment · Area_Chair_cSyi · 2021-08-22
**Please read author response**

Dear reviewers,

Thanks a lot for your review work.

If you haven't done so yet, a friendly reminder to read the author response to your review. Please reply back to the authors and update your score if necessary.

Thanks,
the area chair

---

### Decision · Program_Chairs · 2021-09-27

**Decision:**

Accept (Poster)

**Comment:**

This paper got three 6 ratings and is therefore a borderline accept.

The authors carefully answered reviewers comments and the reviewers were satisfied by the answers.

Moreover, the meta review of the previous ICML submission has been taken into account.

I would therefore like to accept the paper.

The authors should make sure to improve the clarity of the paper for readers without background in topology optimization.